# Polyethylene Glycol-Modified Cationic Liposome as a Promising Nano Spray for Acute Pneumonia Treatment

**DOI:** 10.3390/polym16101384

**Published:** 2024-05-12

**Authors:** Kai Wang, Dagui Chen, Chenxi Zhang, Lu Lu, Fusheng Shang, Yinghua Li

**Affiliations:** 1Key Laboratory of Medical Molecular Virology (MOE/NHC/CAMS), School of Basic Medical Sciences and Shanghai Public Health Clinical Center, Shanghai Medical College, Fudan University, Shanghai 200032, China; kaiwang16@fudan.edu.cn (K.W.); lul@fudan.edu.cn (L.L.); 2Institute of Translational Medicine, Shanghai University, Shanghai 200444, China; dagui1106@shu.edu.cn (D.C.); 18801759639@163.com (C.Z.)

**Keywords:** acute pneumonia, cationic liposomes, polyethylene glycol, ribavirin, inflammation response

## Abstract

Acute pneumonia (AP), triggered primarily by pathogens like bacteria and viruses, is a leading cause of human mortality. Ribavirin, a broad-spectrum antiviral agent, plays a pivotal role in the treatment of AP. However, its therapeutic use is hindered by the need for high dosages and the associated cardiac and hepatic toxicities. In this study, we synthesized polyethylene glycol-modified cationic liposomes to encapsulate ribavirin (RBV-PCL) and formulated it into a spray, aiming to enhance the effectiveness of RBV through respiratory administration. Lipopolysaccharide (LPS), a compound known to induce AP models in animals, was utilized in our research. Successfully, we established an acute pneumonia model in mice using aerosol inhalation. Through animal experiments, we investigated the therapeutic effects of RBV-PCL on mice with AP. In vivo studies revealed promising results. RBV-PCL effectively prolonged the survival of mice with AP, significantly reduced the levels of inflammatory markers such as interleukin-6 (IL-6) and tumor necrosis factor-α (TNF-α), and inhibited the infiltration of neutrophils in the lungs and spleens of mice. These findings suggest that RBV-PCL can effectively suppress the inflammatory response in mice with AP, thus holding significant potential as a novel therapeutic approach for the treatment of acute pneumonia.

## 1. Introduction

Common respiratory infections in humans are primarily caused by viruses, and respiratory viruses spread downward through upper respiratory tract infections to develop acute pneumonia (AP) [1]. AP is an abnormality of gas exchange at the alveolar level accompanied by inflammation of the lung parenchyma, and the infection activates several pattern recognition receptors (PRR) within the alveolar epithelium, eliciting an intense inflammatory response in the lungs to control viral transmission and enhance tissue repair [1]. Ribavirin (RBV) is a monophosphate inosine 5′-monophosphate (IMPDH) inhibitor that inhibits IMPDH, thereby hindering the synthesis of viral nucleic acids. RBV has broad-spectrum antiviral properties, inhibiting a wide range of viruses such as respiratory syncytial virus, influenza virus, herpes simplex virus, and preventing influenza, adenovirus pneumonia, hepatitis A, herpes, and measles [2,3,4]. In China, RBV has been clinically proven to be effective against epidemic hemorrhagic fever, and its efficacy is obvious in early-stage patients, which can reduce the mortality rate, alleviate renal damage, reduce bleeding tendency, and improve systemic symptoms [5]. However, RBV has some common problems, such as a short blood half-life and being easy to be metabolized by the human liver and kidney [3,5]. In addition, unmodified RBV is easily digested, unstable in vivo, and prone to unexpected biological reactions [6]. Therefore, improving the stability of inhibitory RBV for more efficient delivery to intracellular and target tissues has become a key step in the development of inhibitory RBV drugs.

Liposomes (CL) are enclosed vesicles composed of lipid bilayers of phospholipids and cholesterol. With a biofilm-like structure, liposomes can encapsulate both water-soluble and fat-soluble drugs and are highly selective, nontoxic, nonimmunogenic, and suitable for biodegradation, making them the most successful inhalation nanodrug delivery system to date [7,8]. Among the common pulmonary delivery vehicles, polyethylene glycol (PEG) is widely used as a nanomaterial for inhaled pulmonary drug delivery, and hydrophilic and electroneutralized PEG show promise in overcoming the mucus barrier for better drug delivery to infected cells [9,10]. Through PEGylation, drugs are endowed with a hydrophilic layer, altering their hydrophilicity and size to allow them to penetrate the mucus layer. PEG-modified liposomes can reduce the rigidity of the lipid bilayer, provide positive curvature to the lipid bilayer, stabilize the dispersion of liposomes in water by site-blocking, and also enhance the internalization and anticancer effects of PEG-nanoliposome, overcoming the lack of enhanced permeability and retention effect (EPR) action in hematological malignancies and improving therapeutic efficacy [11,12]. Liu developed an azithromycin-containing liposome for the treatment of MRSA infections and found that it did not exhibit significant side effects or toxicity in mice but was able to significantly reduce the bacterial count in a mouse model of MRSA infection and had high antibacterial activity [13].

Chemotherapy administered through the lungs can increase drug concentrations in the lungs and reduce systemic drug exposure. However, inhaled free drugs are highly toxic, tend to be rapidly cleared, and have a nonspecific distribution in the lungs. The use of pulmonary inhalation nanodrug delivery systems for the treatment of lung diseases has many advantages; nanocarriers can encapsulate toxic drugs and release them in a more specific and controlled manner [14,15]. It has been shown that particles obtained by liposome nebulization remain homogeneous and stable and significantly reduce tumor volume in mice [16]. In addition, inhalation of paclitaxel liposomal aerosols had a significant protective effect on rats, reducing alveolitis, alveolar damage, and fibrosis, and thus reducing their mortality [17]. In this study, polyethylene glycol-modified cationic liposomes were prepared and made into spray formulations for effective delivery of RBV to lung cells by respiratory administration. RBV-PCL improved the stability and persistence of RBV in mice, inhibited the inflammatory response induced by lipopolysaccharide (LPS), and showed significant therapeutic effects on AP in mice, making it a promising nanospray formulation for the treatment of AP.

## 2. Materials and Methods

### 2.1. Materials and Instruments

BALB/c female nude mice were purchased from Shanghai Slaughter Laboratory Animal Co. (Shanghai, China). The RAW264.7 and J774.1 cells were purchased from Shanghai FuHeng Biology. Furthermore, 1640 medium, Dulbecco’s modification of Eagle’s medium (DMEM) medium, phosphate buffered saline (PBS), fetal bovine serum, and trypsin were purchased from Gibco. Dichloromethane, ethanol, and chloroform were purchased from Beijing Chemical Industry Group Co. (Beijing, China). Ribavirin (RBV) was purchased from Chongqing Taimeng Pharmaceutical Co. (Chongqing, China). N-hydroxy succinimide (NHS) and 1-(3-dimethylaminopropyl)-3-ethylcarbodiimide (EDC) were purchased from Shanghai Murray Biotechnology Co. (Shanghai, China). Moreover, 4′,6-diamidino-2-phenylindole (DAPI) was purchased from Sigma Co. Fluorescein isothiocyanate (FITC) was purchased from Aladdin (Shanghai, China). The apoptosis kit Alexa fluor 488 was purchased from Invitrogen (Carlsbad, CA, USA). The interleukin-6 (IL-6) and tumor necrosis factor-α (TNF-α) ELISA kits were purchased from Beyotime Co. (Shanghai, China). Hexadecyl-quaternized (carboxymethyl) chitosans (HQCMC), 1,2-dioleoyl-sn-glycero-3-phosphocholine (DOPC), distearoyl phosphatidylethanolamine-polyethylene glycol (DSPE-PEG) [18], cholesterol purchased from Shanghai Yuanye Biotechnology Co. (Shanghai, China). Cell counting kit-8 (CCK8) kit was purchased from Beijing Boao Sen Biotechnology Co. (Shanghai, China). Model BILON-650Y Ultrasonic Cell Breaker was purchased from Shanghai Bilon Instrument Manufacturing Co. (Shanghai, China). Model TR-3L Rotary Evaporator was purchased from Shanghai Taitan Science and Technology Co. (Shanghai, China). Model OM-1500A Spray Dryer was purchased from Shanghai Omeng Industrial Co. (Shanghai, China). Model 939SZ Nanoparticle Size Analyzer was purchased from Auspatronics (Shanghai, China) Co. Model 759S UV–vis spectrophotometer was purchased from Shanghai Prismatic Technology Co. (Shanghai, China). Model 759S UV–vis spectrophotometer was purchased from Shanghai Prism Technology Co. (Shanghai, China). Model HZ200LB Constant Temperature Shaker was purchased from Shanghai Jingsheng Scientific Instrument Co. (Shanghai, China).

### 2.2. Preparation of RBV-PCL

In this study, we referred to the previous report for the preparation of liposomes [19,20]. HQCMC, DOPC, DSPE-PEG, cholesterol, and RBV were co-dissolved in dichloromethane, suspended after sonication (power 27%, 25 °C) for 30 s, and continued to sonicate for 6 min after adding 6 mL of PBS (0.1 mol/L, pH = 7.4). The mixed solution was allowed to undergo an emulsification reaction to obtain a homogeneous emulsion, and dichloromethane was removed by a rotary evaporator to obtain lipid nanoparticles encapsulated with RBV (RBV-PCL). The RBV-PCL lipid suspension was diluted with PBS buffer at pH 7.4, and the inlet temperature was set at 120 °C, the outlet temperature was set at 52 °C, the inlet speed was set at 224 mL/h, and the air velocity was set at 3.4 m/s. Spray drying was carried out in a spray dryer. The inlet and outlet temperatures and vacuum of the cyclone separator were adjusted to make the surface water of the atomized droplets evaporate rapidly. The fully dried particles from the cyclone separator were collected in a collector to obtain RBV-PCL lipid powder aerosol. The obtained powder was sealed and stored in a dryer for testing (Figure 1).

### 2.3. Characterization of the Physicochemical Properties of Nanomaterial Particles

Each group of samples was taken in aliquots of 1 mL, respectively. The particle size and surface zeta potential of each group of samples were determined by a dynamic optical laser particle size analyzer, and the change in particle size of RBV-PCL was detected within 24 h to examine its stability in solution. Moreover, 0.5 mL of the solution was taken and dropped on a copper mesh, and after drying, the morphology and structural characteristics of the samples were observed by scanning electron microscopy (SEM). Fourier transform infrared spectroscopy (FTIR) was used to detect the near-infrared spectra of the nanocarriers.

### 2.4. Drug Release Analysis

The full-wavelength scanning of RBV was carried out by a UV spectrophotometer to determine the UV absorption peak of RBV. Standard solutions of RBV with concentrations of 2, 6, 10, 16, and 20 μg/mL were prepared, and the OD values were determined by UV–visible spectrophotometer, the standard curve was plotted, and the encapsulation rate, drug loading, and drug release profile were determined. Drug loading rate = actual drug concentration/total liposome concentration × 100%. Drug encapsulation rate = actual drug concentration/total drug concentration × 100%. An appropriate amount of RBV-PCL lipid powder aerosol was taken and re-dispersed to 2 mL with PBS (pH = 7.4 and pH = 5.3) so that the drug concentration was about 1 mg/mL and placed in a pre-treated dialysis tubing in a canister containing the release medium. The canister was shaken in a thermostatic shaker with a water bath temperature of (37 ± 0.5) °C and a rotational speed of 50 r/min, and 2 mL of the sample was taken at the set time points of 0 h, 2 h, 4 h, 6 h, 8 h, 10 h, 12 h, 16 h, 20 h, 24 h, 48 h, and 72 h. The UV absorbance was detected, and the release medium at the same thermostatic temperature was added to calculate the release rate of the RBV.

### 2.5. In Vitro Cytotoxicity Assay

An aliquot of 100 μL of the prepared cell suspension of RAW264.7 and J774.1 cells was added to each well of the 96-well plate and incubated at 37 °C in a CO_2_ incubator for 24 h. Different wells were filled with 10 μL of PCL, RBV, and RBV-PCL solutions of different concentrations and incubated at 37 °C in a CO_2_ incubator for 12 h. After 12 h, 10 μL of CCK8 reagent was added to each well and incubated at 37 °C for 4 h, and the OD value was measured at 450 nm (Cell viability % = (OD of administered cell group − OD of blank group)/(OD of control cell group − OD of blank group) × 100%).

### 2.6. Immunofluorescence Experiment

A cell suspension of RAW264.7 and J774.1 (1 × 10^5^) was added to wave bottom petri dishes and incubated for 12 h. Dil was mixed with RBV and vortex-shocked to form Dil-RBV complexes directly by covalent binding of Dil to the hydroxyl group (-OH) in the structural formula of RBV. PCL, fluorescently labeled RBV, and fluorescently labeled RBV-PCL were added to the dishes and incubated at 37 °C in a CO_2_ incubator for 2 h. DAPI and FITC staining solutions were added to each dish, incubated for 20 min under light protection, and tested on the machine.

### 2.7. Enzyme-Linked Immunosorbent Assay Experiment

An aliquot of 100 μL of the prepared cell suspension of RAW264.7 and J774.1 cells was added to each well in a 96-well plate and incubated at 37 °C in a CO_2_ incubator for 24 h. The wells were divided into the control group (NC), PR8 group, PCL/PR8 group, RBV/PR8 group, and RBV-PCL/PR8 group. Among them, all except the NC group were stimulated with LPS, and then different drugs were added according to the grouping and incubated for 12 h at 37 °C in a CO_2_ incubator. The experiments were performed according to the instructions of the ELISA kit, and the results were determined using an enzyme marker.

### 2.8. Mouse Model of AP

The animal study protocol was approved by the Experimental Animal Ethics Committee of Shanghai University, approval No. EAECSHU2023-0189. Moreover, 24 SPF BALB/c mice (six weeks old and female) were divided into NC, PCL, RBV, and RBV-PCL groups, with 6 mice in each group, and the mouse AP model was constructed by aerosol inhalation of LPS. The experimental mice were placed in the glass cylinder of the nebulizer inhaler for drug administration so that they could fully inhale the nebulized drug and were photographed with a small animal in vivo imager at 24 h. The mice in each group were anesthetized with 20% urethane (1 g/kg) and fixed on a frog plate. The chest cavity was then opened, and after exposing the heart, cardiac blood was collected in heparin vacuum blood collection tubes. The number of leukocytes in the blood was determined using a Sysmex KX-21 hematology analyzer (Jinan, China). The trachea was fixed to the cricoid cartilage of mice with arterial forceps, and the right lung was fixed by making an inverted T-shaped metal piece underneath it and placing an intravenous infusion needle attached to a syringe. The left lung was rinsed with 2 mL of saline each time, and the alveolar lavage fluid (BALF) was collected, requiring a recovery rate of 80–90% for 3 times. BALF was centrifuged at 1500 rpm at 4 °C for 10 min, and the supernatant was stored in a −80 °C freezer for spare use. After alveolar lavage, the right lung, heart, liver, kidney, and spleen were removed with eye clippers and eye forceps, respectively. They were placed in a 4% neutral formaldehyde solution for fixation, HE staining, and immunohistochemistry.

### 2.9. Histopathological Analysis

At the end of the experiment, all the lungs of mice in each group were excised and placed in prepared a 4% paraformaldehyde buffer for fixation. Subsequently, they were embedded in paraffin, sectioned 5 µm thick, and stained with hematoxylin and eosin. Histopathological changes were observed under an inverted microscope.

### 2.10. Analysis of Inflammatory Factor Levels

In order to observe the effect of RBV-PCL on inflammatory factors in mice with AP, the mice were divided into four groups, i.e., NC, PCL, RBV, and RBV-PCL. After 1 day of drug administration, alveolar lavage fluid was taken from mice. BALF was used to detect the level of inflammatory factors (IL-6, TNF-α) using an ELISA kit.

### 2.11. Flow Cytometry Analysis

Mouse lungs and spleens were ground and filtered through a cell filter with a pore size of 70 μm. The ground cell suspension was collected and centrifuged at 1500 rpm for 5 min, and the supernatant was discarded. Moreover, 3 mL of pre-cooled 1xACK solution was added, mixed thoroughly to lyse the red blood cells, and left on ice for 5 min. The reaction was then stopped by adding PBS containing 2% serum to 6 mL, centrifuged at 1500 rpm for 5 min, and the supernatant discarded. The cell pellet was resuspended in 1 mL of PBS containing 2% serum and mixed into a single cell suspension for cell counting. The cells were set aside on ice. Take 100 μL of cell suspension, add 1 μL of CD16/CD32 antibody, and incubate for 10 min at room temperature. Furthermore, add the corresponding amount of surface antibody to each tube, mix well, and incubate at room temperature for 20 min. Cells were washed with 1 mL of PBS containing 2% serum, centrifuged at 3000 rpm for 5 min, and the supernatant was discarded. The cells were resuspended in 200 μL of PBS containing 2% serum, filtered through a 300-mesh nylon mesh, and detected by the BD flow cytometry platform.

### 2.12. Statistical Methods

Data were analyzed by one-way ANOVA using GraphPad Prism 9.0.0 statistical software. All data were expressed as the mean ± standard deviation. *p* < 0.05 was considered statistically significant, and *p* < 0.01 was considered highly significant.

## 3. Results

### 3.1. Synthesis and Characterization of RBV-PCL

As shown in Figure 2A,B, the particle size of RBV-PCL was 104.31 ± 7.49 nm, the PDI was 0.164 ± 0.026, and the zeta potential was 35.61 ± 6.58 mV. As the content of RBV drug increased, the particle size continued to increase, and when the mass ratio of RBV drug to PCL was 1:10, the particle size no longer increased and the PDI was less than 0.2 (Figure 2C,D). There was no significant increase in potential with increasing RBV drug content (Figure 2E). FTIR results showed that the absorption peaks of PCL and RBV-PCL were located at 1452–1471 cm^−1^ and 1628–1752 cm^−1^, respectively, corresponding to the absorption peaks caused by methyl and carbonyl stretching, respectively. There was one absorption peak at 2840–2921 cm^−1^ for PCL and two absorption peaks for RBV-PCL, and the absorption peaks were significantly enhanced, indicating that the drug had successfully encapsulated PCL (Figure 2F). The results of the SEM analysis showed that RBV-PCL had a spherical vesicle structure with good dispersion and no agglomeration (Figure 2G).

### 3.2. Drug Loading Efficiency and Drug Release Performance Assay

The UV scanning curve showed that RBV had a distinct UV absorption peak at 207 nm (Figure 3A). The standard curve of RBV was plotted by UV: y = 0.0439x + 0.176, R2 = 0.9999 (Figure 3B). With the increase in RBV drug content, the encapsulation efficiency gradually decreased, and the drug loading gradually increased. Therefore, the best encapsulation efficiency and drug loading were achieved when the mass ratio of RBV to PCL was 1:10, with the encapsulation efficiency and drug loading being (95.25 ± 4.48) % and (10.04 ± 0.26) %, respectively (Figure 3C,D). In the acidic environment at pH = 5.4, the cumulative release rate of RBV reached more than 80% within 10 h, while the cumulative release rate of RBV-PCL was (80 ± 2.84) % within 48 h (Figure 3E). At the release medium pH = 7.4, the cumulative release rate of RBV reached more than 80% within 8 h, while the cumulative release rate of RBV-PCL was (86 ± 2.53) % within 48 h (Figure 3F). It indicated that RBV-PCL possessed significant slow-release properties, and the drug release rate was significantly accelerated in a neutral environment.

### 3.3. RBV-PCL Cytotoxicity, Uptake, and Effect on Secretion of the Inflammatory Factor IL-6 by LPS-Stimulated Cells

The results of CCK-8 are shown in Figure 4A,B. There were no obvious cytotoxic side effects of RBV, PCL, and RBV-PCL in all concentration ranges, and the cell survival rate was greater than 80%, so the concentration of 100 μg/mL was chosen for the experiment. IL-6 is a relatively common inflammatory factor, and LPS can stimulate RAW264.7 and J774.1 cells to secrete high levels of IL-6. Compared with the PR8 group, RBV and RBV-PCL significantly reduced the level of IL-6 secreted by RAW264.7 and J774.1 cells, indicating that RBV-PCL has a significant inhibitory effect on the inflammatory factors secreted by RAW264.7 and J774.1 cells stimulated by LPS (Figure 4C,D). As shown in Figure 4E,F, only sporadic weak red light was observed in the cells of the RBV group, indicating that free RBV could not enter the cells efficiently and that most of the RBV was only adsorbed on the membrane surface. In contrast, RBV-PCL showed higher intracellular delivery efficiency, with more RBV entering the cell and even the nucleus. These results suggest that PCL liposomes have the potential to be developed as RBV delivery carriers.

### 3.4. RBV-PCL Inhibits LPS-Induced Acute Lung Injury in Mice In Vivo

As shown in Figure 5A, RBV has a small distribution range and low fluorescence intensity in vivo, which may be related to the fast metabolism in the blood. The distribution range and fluorescence intensity of the RBV-PCL group were higher than those of the RBV group. These results suggest that PCL can effectively reduce RBV clearance in vivo and maintain a high level of immunosuppression. The survival rate of the AP mouse model showed that the survival rate of the AP mice started to die on the 5th day after infection, and the survival rate of mice in the RBV and RBV-PCL groups was 45% and 62%, respectively, which indicated that both RBV and RBV-PCL reduced the mortality rate of AP mice, but the effect of RBV-PCL was more obvious (Figure 5B). The results of Figure 5C,D showed that the lung tissues of the control group were structurally normal and no histopathological changes were observed. The lung tissue of mice in the PCL group was dark, with massive hemorrhage in the lobes and alveoli, interstitial edema, and massive inflammatory cell infiltration in the lobes. Compared with the PCL group, mice in the RBV group showed reduced lung tissue damage, but the alveolar walls were still thickened, and a small portion of the tissue was congested with inflammatory cell infiltration. In contrast, the histological damage in the RBV-PCL group was significantly reduced, with essentially normal non-histological structures and only a small amount of inflammatory cell infiltration. Pathological scores also showed that RBV-PCL had a more pronounced effect on the reduction of inflammatory injury in mice with AP.

### 3.5. Effects of RBV-PCL on the Levels of Inflammatory Factors and Infiltration of Neutrophils and Macrophages in LPS-Infected Mice

As shown in Figure 6A,B, IL-6 and TNF-α levels decreased in both the RBV and RBV-PCL groups compared with the PCL group, but the decrease was more pronounced in the RBV-PCL group. It indicates that RBV-PCL more significantly inhibited IL-6 and TNF-α levels in the cysteine of LPS-stimulated mice. As shown in Figure 6C,D, the proportions of splenic neutrophils in mice in the NC, PCL, RBV, and RBV-PCL groups were 2.2%, 14.8%, 6.58%, and 4.35%, respectively. The proportions of neutrophils in the lungs of mice in the NC, PCL, and RBV-PCL groups were 8.32%, 58.89%, 25.6%, and 22.6%, respectively. It indicates that RBV-PCL can regulate the changes in neutrophils in mice to achieve the effect of inhibiting inflammation. Figure 6E,F showed that the number of M1 macrophages significantly increased and the number of M2 macrophages significantly increased and decreased in the lungs of mice after LPS stimulation (*p* < 0.05). The number of M1 macrophages decreased to 22.6% and 18.5%, and the number of M2 macrophages increased to 23.5% and 52.6% in the RBV and RBV-PCL groups (*p* < 0.001). The results indicated that RBV-PCL further reduced the number of M1 macrophages and increased the number of M2 macrophages in mouse lungs, thereby suppressing inflammation.

## 4. Discussion

Most of the major causes of pathogenic bacterial infections leading to severe damage and even death of the host organism are uncontrollable excessive inflammatory responses. Cytokine storms have been reported to be associated with the worsening of many infectious diseases, such as severe acute respiratory syndrome (SARS), Middle East respiratory syndrome (MERS), and the novel coronavirus pandemic of 2019 [19]. Ribavirin inhibits both DNA and RNA viruses. In vitro studies have shown that ribavirin enhances the in vitro activity of interferon, and that ribavirin can be effective in treating patients with SARS-CoV infection when administered immediately after diagnosis [20,21]. However, the pulmonary barrier limits the range of drugs that can be used for direct pulmonary inhalation, and most drugs cannot be delivered directly to the lungs due to their inherent limitations, such as superhydrophobicity and poor cellular uptake [22]. Functionalized biomimetic materials offer a novel approach for targeted therapy. Our previous research has shown that metal–organic frameworks (MOFs) are promising delivery carriers for small molecules and nucleic acid drugs [23,24]. MOFs have been reported as a viable option for nebulization therapy in the throat due to the effective deposition of micrometer-sized MOF particles within the throat region [25]. Recent advancements in cell membrane-mediated drug delivery have the potential to enhance the efficacy of AP treatment by precisely targeting potent corticosteroid drugs to hyper-activated immune cells [26]. Liposomes, due to their similar cellular structure and passive targeting, are often used as drug carriers for pulmonary drug delivery [27]. Compared to MOF and cell membranes, liposomes are easier to prepare and avoid the threat of metal ions, which facilitate the direct delivery of antiviral drugs and act on lung cells to inhibit viral reproduction.

PEG has the ability to protect nucleic acids from nuclease degradation and is widely used for in vivo and in vitro siRNA and DNA delivery due to its “proton sponge effect” and its ability to escape efficiently in vivo [28,29,30]. It has now become one of the gold standards for non-viral gene delivery. Delivery of siRNA into cancer cells using PEG liposomes has demonstrated that PEG liposomes can be used as a safe and effective delivery vehicle [31,32]. Togami encapsulated bleomycin in liposomes and administered it to the lungs of mice, finding that it significantly increased the anticancer activity of the drug, reduced adverse effects, and did not cause pharmacologic lung injury [33]. Fu investigated the safety of nebulized inhaled IFN-κ + TFF2 liposomes in patients with novel coronaviruses and observed no significant toxicity and significantly promoted clinical improvement and viral RNA reversal [34]. Effective pulmonary inhaled drug delivery requires overcoming relevant barriers, including anatomical, physical, immunologic, and metabolic barriers. Due to physiological limitations of the lungs, inhalation of agents with particle sizes smaller than 1~3 μm is required to penetrate deeply into the alveoli. The interaction of nanoparticles with various biological barriers is highly dependent on the physicochemical properties of the nanoparticles, including size, shape, surface charge, and surface hydrophobicity. For example, nanoparticles with sizes less than 100 nm have shown superior ability to overcome the site barrier of mucus and EPS, allowing sufficient mucus and biofilm penetration [35,36].

The physicochemical properties of RBV-PCL prepared in this study were satisfactory. RBV-PCL had good dispersion and stability with a particle size of 104.31 ± 7.49 nm, was spherical in shape, and was extensively bound to RBV by charge adsorption. The results of laser confocal microscopy showed that PCL could effectively deliver RBV into cells with stronger delivery efficiency than RBV alone. In order to further improve the anti-inflammatory effect of RBV and to address the drawback of the poor permeability of individual RBV drugs to cell membranes, liposomes were used for intracellular drug delivery in this study. The results showed that PCL could effectively promote the entry of RBV into RAW264.7 and J774.1 cells and effectively reduce the inflammatory response induced by LPS stimulation. In the in vivo experiments, the survival rate of mice in the RBV-PCL group was significantly higher than that in the RBV-alone group, and the degree of pneumonia injury and the level of inflammatory factor in alveolar lavage fluid were lower than those in the RBV-alone group. In the in vivo experiments, the drug action time was prolonged, and the inclusiveness of the animal body was strong, so that the drug could work better.

Neutrophils are a class of leukocytes that play an important role in the acute inflammatory response and are the body’s first line of cellular defense against pathogens. Mature neutrophils generally have a short half-life in the somatic circulation and die rapidly through apoptosis. Inhibition of neutrophil apoptosis has been shown to exacerbate inflammatory responses [37,38]. In this study, we observed by flow cytometry that RBV-PCL reduced the number of neutrophils in the spleen and lung of mice with AP. M1 macrophages, also known as “inflammatory” macrophages, are rapidly produced in response to collective stimuli. Apoptosis of macrophages has been shown to promote the resolution of inflammatory responses in vivo [39,40]. M2 macrophages have anti-inflammatory and tissue-regenerative effects that are generally induced by IL-4 and IL-13. In this study, we observed by flow cytometry that RBV-PCL decreased the number of M1 macrophages and increased the number of M2 macrophages in the lungs of mice with AP. This also suggests that RBV-PCL can control the progression of inflammation by regulating the number of inflammatory cells in vivo.

## 5. Conclusions

In this study, PEG-modified cationic liposomes were selected as drug delivery carriers to effectively deliver RBV into tissues and cells by respiratory administration, which improved the stability and persistence of RBV in mouse lungs, inhibited the inflammatory response induced by influenza virus in vivo and in vitro, thereby prolonging the survival rate of the mice, and attenuated the lung injury and the level of inflammatory factors in the mouse model of viral pneumonia. This study suggests that PCL liposomes are expected to be effective drug delivery vehicles, and RBV-PCL is also expected to be a potential drug for the treatment of immune-mediated inflammatory diseases.

## Figures and Tables

**Figure 1 polymers-16-01384-f001:**
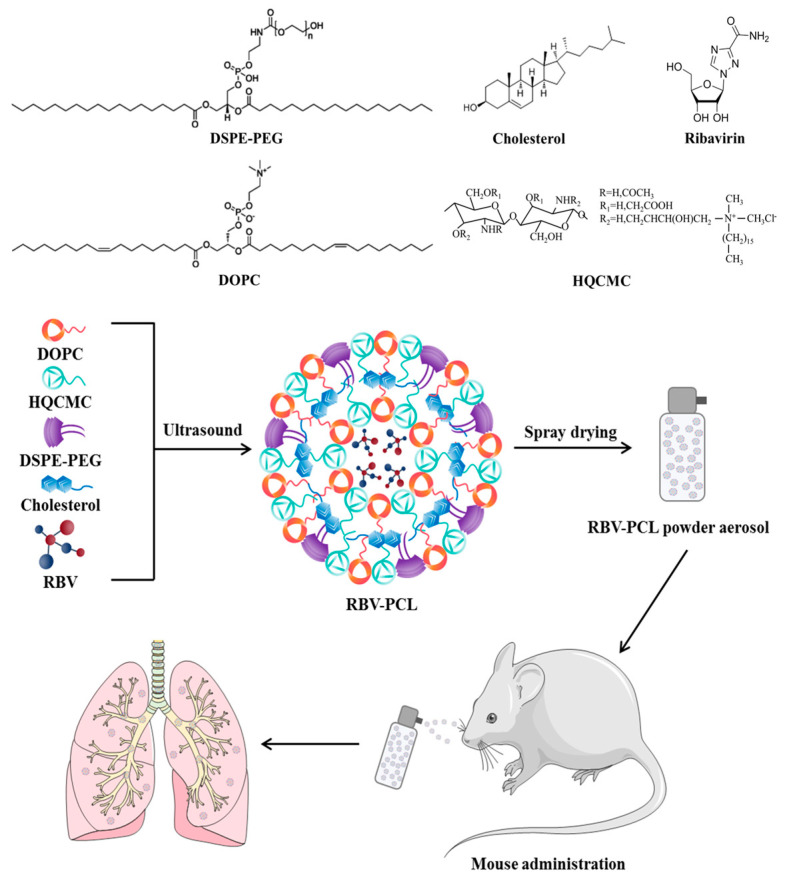
Compound structural formula, preparation process of nanoliposomes, and animal experimentation process.

**Figure 2 polymers-16-01384-f002:**
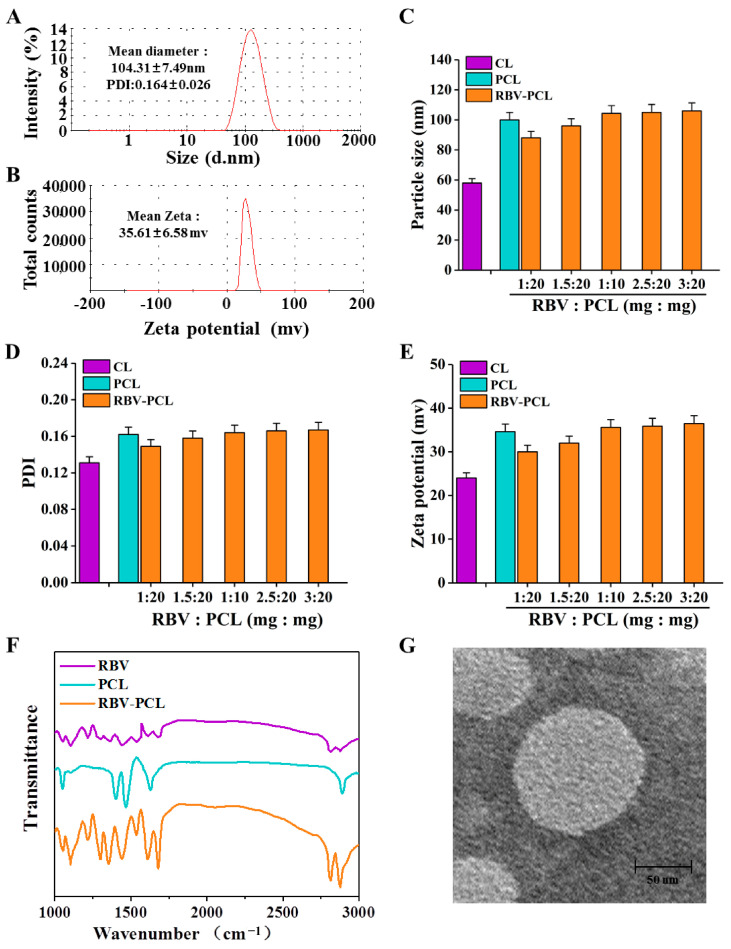
Analysis of physicochemical characterization tests of liposomes. (**A**) Particle size test of RBV-PCL. (**B**) Potential test of RBV-PCL. (**C**) Particle size test of different liposomes. (**D**) PDI of different liposomes. (**E**) Potentiometric test of different liposomes. (**F**) FTIR test of RBV-PCL. (**G**) SEM of RBV-PCL.

**Figure 3 polymers-16-01384-f003:**
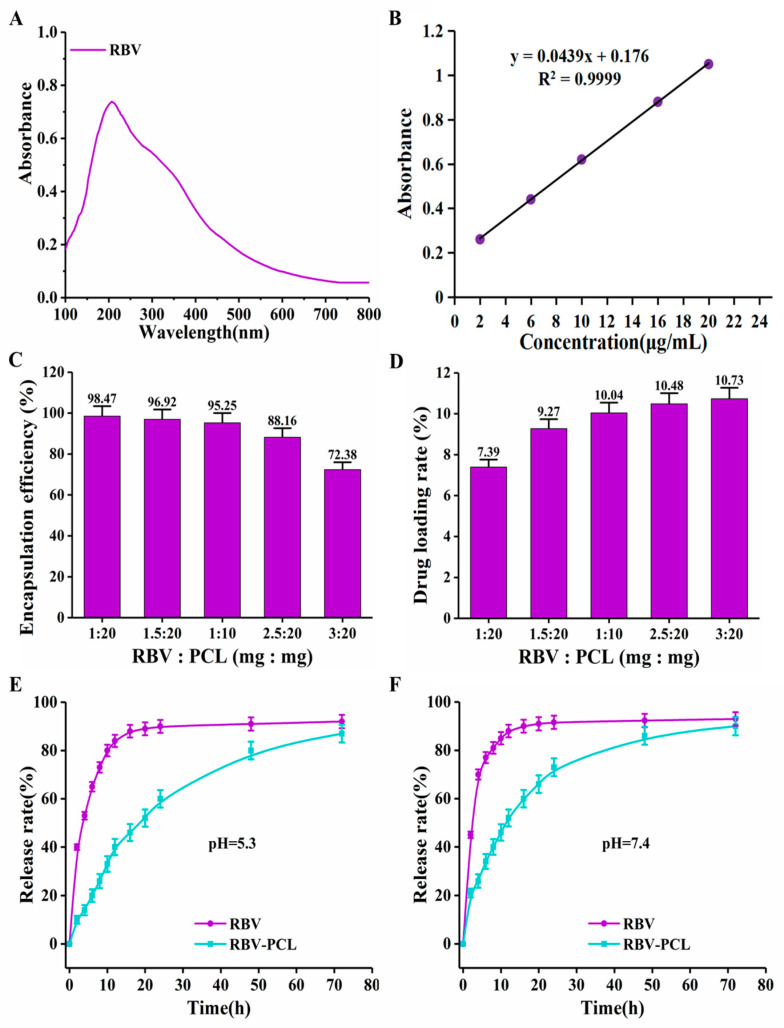
Loading performance and drug release performance assays of RBV-PCL. (**A**) UV scanning curve. (**B**) Standard curve of RBV. (**C**) Encapsulation rate test after adding different contents of RBV drugs to RBV-PCL. (**D**) Drug loading capacity test after adding different contents of RBV drug in RBV-PCL. (**E**) Drug release profiles of RBV and RBV-PCL at pH = 5.3. (**F**) Drug release profiles of RBV and RBV-PCL at pH = 7.4.

**Figure 4 polymers-16-01384-f004:**
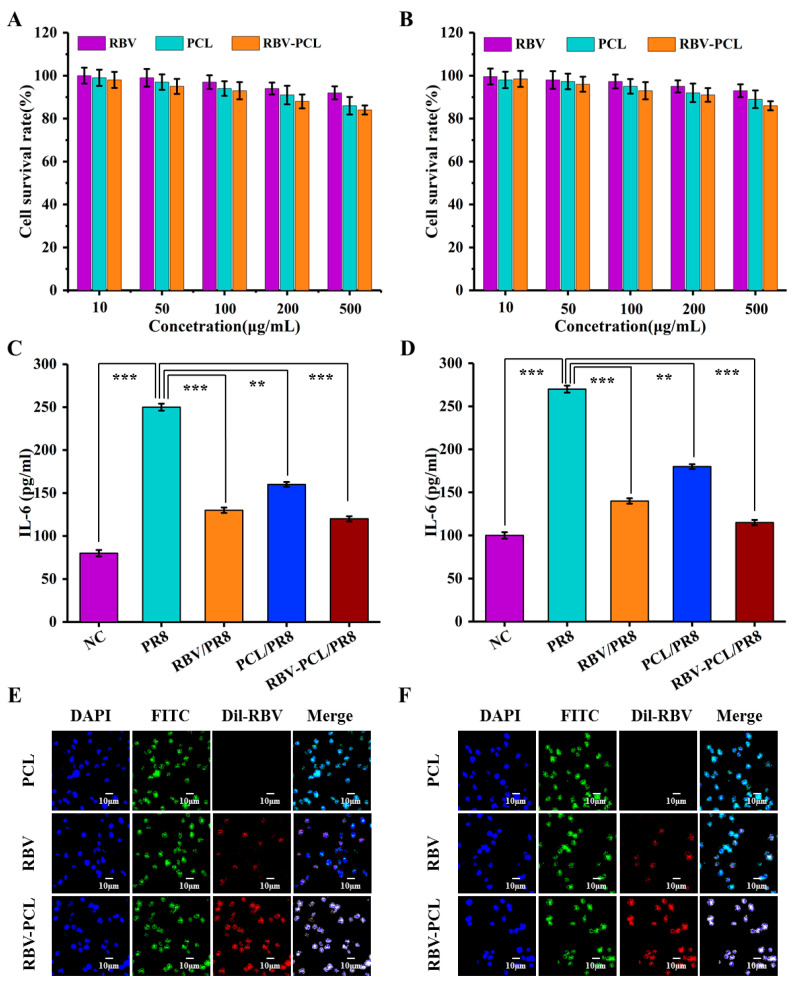
Toxicity, proliferation, and apoptosis assays of nanoliposomes on cells. (**A**) RAW264.7 cytotoxicity assay. (**B**) J774.1 cytotoxicity assay. (**C**) Effect of RBV-PCL on IL-6 secretion by RAW264.7 cells. (**D**) Effect of RBV-PCL on IL-6 secretion by J774.1 cells. (**E**) RAW264.7 cell uptake assay. (**F**) J774.1 cell uptake experiments. ** *p* < 0.01 and *** *p* < 0.001.

**Figure 5 polymers-16-01384-f005:**
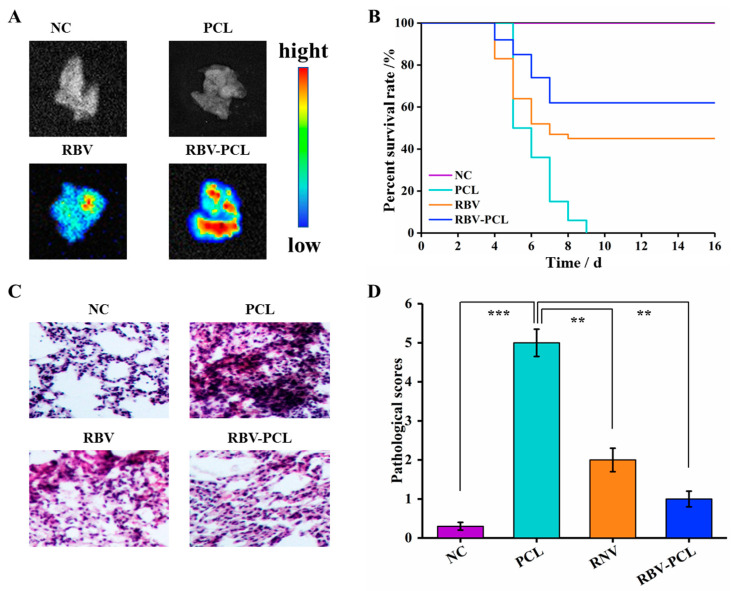
Effect of RBV-PCL on lung injury in vivo. (**A**) Small animal in vivo imaging system to track the distribution of RBV-PCL in mice. (**B**) Effect of RBV-PCL on survival of mice with AP. (**C**) HE staining to observe the histopathological changes in the lungs of mice. (**D**) Pathological scoring of histopathological sections of the lungs of mice. ** *p* < 0.01 and *** *p* < 0.001.

**Figure 6 polymers-16-01384-f006:**
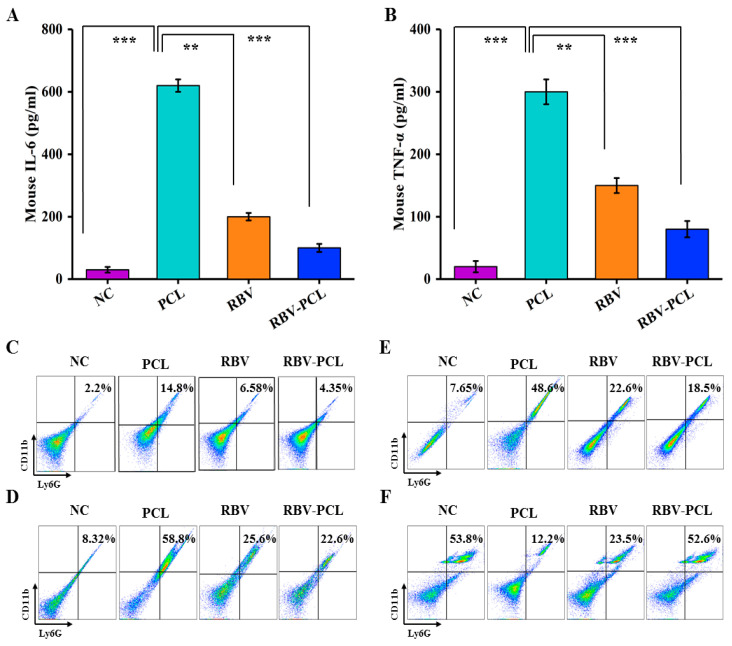
Inflammatory factor levels and inflammatory cell assays in mouse alveolar lavage fluid. (**A**) IL-6 levels in mouse alveolar lavage fluid. (**B**) TNF-α levels in mouse alveolar lavage fluid. (**C**) flow cytometric analysis of mouse spleen. (**D**) mouse lung flow cytometric analysis. (**E**) mouse lung macrophage M1 flow cytometric analysis. (**F**) Mouse lung macrophage M2 flow cytometric analysis. ** *p* < 0.01 and *** *p* < 0.001.

## Data Availability

Data are contained within the article.

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
