# Peer review of "Polyethylene Glycol-Modified Cationic Liposome as a Promising Nano Spray for Acute Pneumonia Treatment"

_polymers, 2024, doi:10.3390/polym16101384_

Round 1

Reviewer 1 Report

Comments and Suggestions for Authors

The manuscript polymers-2975169 “Polyethylene Glycol-modified Cationic Liposome as a Promising Nano Spray for Acute Pneumonia Treatment” by Kai Wang et al. describes the development of polyethylene glycol modified cationic liposomes for encapsulating ribavirin with improved pharmacological activity for inhalation administration. 

The topic of the article is interesting, but the manuscript needs significant revision. Methodologies should be rewritten clearly and concisely. Without a clear understanding of the experimental methodology, it is difficult to evaluate the obtained results. The English should also be improved.

Questions and comments:

  1. Lines 15-16 - Lipopolysaccharide is better called a compound rather than a drug.

  2. Lines 33 -35 - The sentence is difficult to understand. It should be revised.

  3. Line 56 - The abbreviation EPR should be explained.

  4. Case studies of the use of liposomes for inhalation administration of antibiotics should be added to the Introduction.

  5. Check all abbreviations in the text. They should be deciphered at first mention.

  6. Line 91 - HQCMC,DOPC,DSPE-PEG - Need to specify what this is and how it is obtained.

  7. Sections 2.2, 2.3 and 2.4. - The used instruments should be specified.

  8. Section 2.4. - What does "Actual drug loading rate" mean? 

  9. What is CCK8 reagent? Add it to Section 2.1.

  10. Section 2.1. - Not all the specified reagents are used in further experiments. Check that.

  11. It is not clear whether the MTT test was used to determine cytotoxicity.

  12. Add a fluorescence labeling technique for the samples. 

  13. Experiments of anti-inflammatory activity should be described in more detail.What markers of inflammation have been studied?

  14. Due to what liposomes have a positive charge?

  15. Line 319 - PEG is not a cationic polymer.

Author Response

Dear Reviewers:

Thank you for your letter and for the reviewers’ comments concerning our manuscript entitled “Polyethylene Glycol-modified Cationic Liposome as a Promising Nano Spray for Acute Pneumonia Treatment” (ID: polymers-2975169). Those comments are all valuable and very helpful for revising and improving our paper, as well as the important guiding significance to our researches. We have studied comments carefully and have made correction which we hope meet with approval. Revised portion are marked in red in the paper. The main corrections in the paper and the responds to the reviewer’s comments are as flowing:

Comments and Suggestions for Authors

The manuscript describes the development of polyethylene glycol modified cationic liposomes for encapsulating ribavirin with improved pharmacological activity for inhalation administration. 

The topic of the article is interesting, but the manuscript needs significant revision. Methodologies should be rewritten clearly and concisely. Without a clear understanding of the experimental methodology, it is difficult to evaluate the obtained results. The English should also be improved.

Questions and comments:

  1. Lines 15-16 - Lipopolysaccharide is better called a compound rather than a drug.

Reply: Response: Thank you very much for your suggestion. In the revised draft, we have changed "drug" to "compound".

  1. Lines 33 -35 - The sentence is difficult to understand. It should be revised.

Reply: Thank you very much for your suggestion. In the revised draft, we have changed " Ribavirin (RBV) is a monophosphate hypoxanthine nucleoside dehydrogenase inhibitor (IMP) that inhibits IMP, thus hindering the synthesis of viral nucleic acids." to "Ribavirin (RBV) is a monophosphate inosine 5′-monophosphate dehydrogenase (IMPDH) inhibitor that inhibits IMPDH, thereby hindering the synthesis of viral nucleic acids.".

  1. Line 56 - The abbreviation EPR should be explained.

Reply: Thank you very much for your suggestion. In the revised version, we add the full name of EPR. Specifically, "enhanced permeability and retention effect (EPR)".

  1. Case studies of the use of liposomes for inhalation administration of antibiotics should be added to the Introduction.

Reply: Thank you very much for your suggestion, in the revised draft we have added a description of studies related to inhalation drug delivery using liposomes in the introduction. Specifically " Liu developed an azithromycin-containing liposome for the treatment of MRSA infections and found that it did not exhibit significant side effects or toxicity in mice, but was able to significantly reduce the bacterial count in a mouse model of MRSA infection and had high antibacterial activity [13].".

  1. Check all abbreviations in the text. They should be deciphered at first mention.

Reply: Thank you very much for your suggestion. In the revised version, we have added the full name of the abbreviation when it first appears. For example, " interleukin- 6 (IL-6), tumor necrosis factor-α (TNF-α), lipopolysaccharide (LPS), dulbecco's modification of eagle's medium dulbecco (DMEM), Hexadecyl-quaternized (carboxymethyl) Chitosans (HQCMC), 1,2-dioleoyl-sn-glycero-3-phosphocholine (DOPC), Distearoyl phosphatidylethanolamine-polyethylene glycol (DSPE-PEG), etc.".

  1. Line 91 - HQCMC, DOPC, DSPE-PEG - Need to specify what this is and how it is obtained.

Reply: Thank you very much for your suggestions. In the revised version, we have added the sources of these materials to the reagent materials.

  1. Sections 2.2, 2.3 and 2.4. - The used instruments should be specified.

Reply: Thank you very much for your suggestion. In the revised draft, we have listed the models and sources of the necessary instruments. Specifically " Model BILON-650Y Ultrasonic Cell Breaker was purchased from Shanghai Bilon In-strument Manufacturing Co. Model TR-3L Rotary Evaporator was purchased from Shanghai Taitan Science and Technology Co. Model OM-1500A Spray Dryer was pur-chased from Shanghai Omeng Industrial Co. Model 939SZ Nano Particle Size Analyzer was purchased from Auspatronics (Shanghai) Co. Model 759S UV-Vis Spectrophotometer was purchased from Shanghai Prismatic Technology Co. Model 759S UV-Vis Spectrophotometer was purchased from Shanghai Prism Technology Co. Model HZ200LB Constant Temperature Shaker was purchased from Shanghai Jingsheng Scientific Instrument Co.".

  1. Section 2.4. - What does "Actual drug loading rate" mean?

Reply: Thank you very much for your advice.  "Actual drug loading rate" refers to the amount of drug contained in the package.  In the revised draft, we have changed it to "Drug loading rate".

  1. What is CCK8 reagent? Add it to Section 2.1.

Reply: Thank you very much for your suggestion. The full name of CCK8 is Cell Counting Kit-8 and in the revision we have added CCK8 to section 2.1. Specifically, "Cell counting kit-8 (CCK8) kit was purchased from Beijing Boao Sen Biotechnology Co.".

  1. Section 2.1. - Not all the specified reagents are used in further experiments. Check that.

Reply: Thank you very much for your suggestion. In the revised version, we have deleted "Propionate, mPEG2000-OH, stannous octanoate and ethylenate were purchased from Sigma." and "MTT kit was purchased from Beijing Chemical Industry Co. ".

  1. It is not clear whether the MTT test was used to determine cytotoxicity.

Reply: Thank you very much for your comments. We are using CCK8 is used to detect cytotoxicity, and in the revised draft, we have removed the mischaracterization.

  1. Add a fluorescence labeling technique for the samples.

Reply: Thank you very much for your suggestion. In the revised version, we have added the labeling method. Specifically, "Dil was mixed with RBV and vortex-shocked to form Dil-RBV complexes directly by covalent binding of Dil to the hydroxyl group (-OH) in the structural formula of RBV.".

  1. Experiments of anti-inflammatory activity should be described in more detail.What markers of inflammation have been studied?

Reply: Thank you very much for your suggestion. We have added inflammation indicators to our methodology. Specifically, "IL-6, TNF-α".

  1. Due to what liposomes have a positive charge?

Reply: Thank you very much for your query. We have incorporated Hexadecyl-quaternized (carboxymethyl) chitosans (HQCMC) in our material. Chitosan is a natural cationic polysaccharide, and it is possible to prepare cationic liposomes using chitosan as a matrix material.

  1. Line 319 - PEG is not a cationic polymer.

Reply: Thank you very much for your suggestion. In the revised version, we have removed the incorrect description. We have changed "PEG is not a cationic polymer " to "PEG has".

Reviewer 2 Report

Comments and Suggestions for Authors

In this study, authors have developed the spray formulation of polyethylene glycol-modified cationic liposomes which loaded with ribavirin for acute pneumonia management through respiratory administration. An acceptable survival in acute pneumonia induced mice was observed accompany with reduced inflammation response upon decreases IL-6 and TNF-α levels.

The manuscript is valuable and written well. However, authors should address the issues.

·         How authors explain the stability of RBV in RBV-PCL form?

·         There are some grammatical mistake in the text that must be corrected. For example in line space between word and parenthesis is needed.

……………….sonication(Power 27%, ……..

·         Authors should specify the abbreviations of HQCMC,DOPC,DSPE in the text for the first time.

·         Authors should mention the ethical statement and approval code of the related committee for the animal study. The number of animals for each group should be explained.

·         In Fig.2F, please present the FTIR spectrum of pure RBV for comparison.

·         Authors should explain why the release of RBV from the formulation is not pH sensitive.

·         Authors should present the quantitative results of immune cell infiltration in H&E investigation.

·         How authors have measure the effectiveness of the RBV PCL onthe acute and chronic state of the acute pneumonia.

·         Authors should compare their finding with the similar studies and highlight the advantages of the study.

·         The conclusion should revised again and extended more.

Best

Comments on the Quality of English Language

 Minor editing of English language required.

Author Response

Dear Reviewer:

Thank you for your letter and for the reviewers’ comments concerning our manuscript entitled “Polyethylene Glycol-modified Cationic Liposome as a Promising Nano Spray for Acute Pneumonia Treatment” (ID: polymers-2975169). Those comments are all valuable and very helpful for revising and improving our paper, as well as the important guiding significance to our researches. We have studied comments carefully and have made correction which we hope meet with approval. Revised portion are marked in red in the paper. The main corrections in the paper and the responds to the reviewer’s comments are as flowing:

Comments and Suggestions for Authors

In this study, authors have developed the spray formulation of polyethylene glycol-modified cationic liposomes which loaded with ribavirin for acute pneumonia management through respiratory administration. An acceptable survival in acute pneumonia induced mice was observed accompany with reduced inflammation response upon decreases IL-6 and TNF-α levels.

The manuscript is valuable and written well. However, authors should address the issues.

  1. How authors explain the stability of RBV in RBV-PCL form?

Reply: Thank you very much for your suggestion. In the original manuscript, we examined the change in particle size of RBV-PCL over 24h to investigate its stability in solution. It was found that the particle size continued to increase with the increase of RBV drug content, and when the mass ratio of RBV drug to PCL was 1:10, the particle size no longer increased, the PDI was less than 0.2, and there was no significant increase in the potential (Fig. 2C-E), which indicated that the stability of RBV-PCL was good. In addition, we were unable to assess the stability of RBV in the RBV-PCL form, since RBV is slowly released in the RBV-PCL form.

  1. There are some grammatical mistakes in the text that must be corrected. For example in line space between word and parenthesis is needed.

………………. sonication (Power 27%, ……..

Reply: Thank you very much for your suggestions. In the revised draft, we have corrected these errors.

  1. Authors should specify the abbreviations of HQCMC, DOPC, DSPE in the text for the first time.

Thank you very much for your suggestion. In the revised draft, we have indicated the full name of HQCMC, DOPC, DSPE. Specifically, " Hexadecyl-quaternized (carboxymethyl) Chitosans (HQCMC), 1,2-dioleoyl-sn-glycero-3-phosphocholine (DOPC), Distearoyl phosphatidylethanolamine-polyethylene glycol (DSPE-PEG)".

  1. Authors should mention the ethical statement and approval code of the related committee for the animal study. The number of animals for each group should be explained.

Reply: Thank you very much for your advice. In the revised draft, we have added the ethical number. Specifically, " EAECSHU 2023-0189". In addition, we added the number of animals, specifically "Twenty-four BALB/c mice were divided into NC, PCL, RBV and RBV-PCL groups, with 6 mice in each group,".

  1. In Fig.2F, please present the FTIR spectrum of pure RBV for comparison.

Reply: Thank you very much for your suggestion. In the revised version, we have added the FTIR spectrum of RBV.

  1. Authors should explain why the release of RBV from the formulation is not pH sensitive.

Reply: The RBV-PCL prepared in this study was prepared from dioleoylcholine (DOPC), cholesterol, etc., and the liposomes prepared with DOPC as a matrix material were less sensitive to pH compared to phospholipid materials such as dioleoylphosphatidylethanolamine.

  1. Authors should present the quantitative results of immune cell infiltration in H&E investigation.

Reply: Thank you very much for your suggestion. In the original manuscript, we have scored the HE pathology staining results. See Figure 5D.

  1. How authors have measured the effectiveness of the RBV PCL on the acute and chronic state of the acute pneumonia.

Reply: Thank you very much for your suggestion. In this study, we did not evaluate the efficacy of RBV-PCL in the acute and chronic stages of acute pneumonia. These suggestions will be invaluable to us in future in-depth studies.

  1. Authors should compare their finding with the similar studies and highlight the advantages of the study.

Reply: Thank you very much for your suggestion. In the revised version, we have added a description of the results of the relevant studies. Specifically, " Liu developed an azithromycin-containing liposome for the treatment of MRSA infections and found that it did not exhibit significant side effects or toxicity in mice, but was able to significantly reduce the bacterial count in a mouse model of MRSA infection and had high antibacterial activity [13]".

  1. The conclusion should revise again and extended more.

Reply: Thank you very much for your suggestion, and in the revised version we have rewritten the conclusion. Specifically, "In this study, PEG-modified cationic liposomes were selected as drug delivery carriers to effectively deliver RBV into tissues and cells by respiratory administration, which improved the stability and persistence of RBV in mouse lungs, inhibited the inflammatory response induced by influenza virus in vivo and in vitro, thereby prolonging the survival rate of the mice, and attenuating the lung injury and the level of inflammatory factors in the mouse model of viral pneumonia. This study suggests that PCL liposomes are expected to be effective drug delivery vehicles and RBV-PCL is also expected to be a potential drug for the treatment of immune-mediated inflammatory diseases.".

Round 2

Reviewer 1 Report

Comments and Suggestions for Authors

The reviewer's comments have been addressed. The manuscript may be accepted.